# Aging and Caloric Restriction Modulate the DNA Methylation Profile of the Ribosomal RNA Locus in Human and Rat Liver

**DOI:** 10.3390/nu12020277

**Published:** 2020-01-21

**Authors:** Noémie Gensous, Francesco Ravaioli, Chiara Pirazzini, Roberto Gramignoli, Ewa Ellis, Gianluca Storci, Miriam Capri, Stephen Strom, Ezio Laconi, Claudio Franceschi, Paolo Garagnani, Fabio Marongiu, Maria Giulia Bacalini

**Affiliations:** 1Department of Experimental, Diagnostic and Specialty Medicine, Alma Mater Studiorum, 40138 Bologna, Italy; noemieelise.gensous2@unibo.it (N.G.); francesco.ravaioli2@unibo.it (F.R.); gianluca.storci@gmail.com (G.S.); miriam.capri@unibo.it (M.C.); claudio.franceschi@unibo.it (C.F.); 2IRCCS Istituto delle Scienze Neurologiche di Bologna, 40 139 Bologna, Italy; chiara.pirazzini@ausl.bologna.it (C.P.); mariagiulia.bacalini@ausl.bologna.it (M.G.B.); 3Department of Laboratory Medicine, Division of Pathology, Karolinska Institutet, 171 76 Stockholm, Sweden; roberto.gramignoli@ki.se; 4Department of Clinical Science, Intervention and Technology (CLINTEC), Karolinska Institutet, 141-86 Stockholm, Sweden; ewa.ellis@ki.se; 5Department of Laboratory Medicine, Karolinska Institute and Karolinska Universitetssjukhuset, 171 76 Stockholm, Sweden; stephen.strom@ki.se; 6Department of Biomedical Sciences, University of Cagliari, 09 124 Cagliari, Italy; elaconi@unica.it (E.L.); fabiomarongiu@unica.it (F.M.); 7Laboratory of Systems Medicine of Healthy Aging and Department of Applied Mathematics, Lobachevsky Univeristy, 603950 Nizhny Novgorod, Russia; 8Department of Laboratory Medicine, Clinical Chemistry, Karolinska Institutet, Karolinska University Hospital, 171 76 Stockholm, Sweden; 9Applied Biomedical Research Center (CRBA), Policlinico S. Orsola-Malpighi Polyclinic, 40 138 Bologna, Italy; 10CNR Institute of Molecular Genetics “Luigi Luca Cavalli-Sforza”, Unit of Bologna, 40136 Bologna, Italy

**Keywords:** aging, DNA methylation, liver, ribosomal RNA, caloric restriction

## Abstract

A growing amount of evidence suggests that the downregulation of protein synthesis is an adaptive response during physiological aging, which positively contributes to longevity and can be modulated by nutritional interventions like caloric restriction (CR). The expression of ribosomal RNA (rRNA) is one of the main determinants of translational rate, and epigenetic modifications finely contribute to its regulation. Previous reports suggest that hypermethylation of ribosomal DNA (rDNA) locus occurs with aging, although with some species- and tissue- specificity. In the present study, we experimentally measured DNA methylation of three regions (the promoter, the 5′ of the 18S and the 5′ of 28S sequences) in the rDNA locus in liver tissues from rats at two, four, 10, and 18 months. We confirm previous findings, showing age-related hypermethylation, and describe, for the first time, that this gain in methylation also occurs in human hepatocytes. Furthermore, we show that age-related hypermethylation is enhanced in livers of rat upon CR at two and 10 months, and that at two months a trend towards the reduction of rRNA expression occurs. Collectively, our results suggest that CR modulates age-related regulation of methylation at the rDNA locus, thus providing an epigenetic readout of the pro-longevity effects of CR.

## 1. Introduction

A critical ability of living beings is to maintain energy homeostasis in response to variations in environmental conditions. These variations occur not only in the “external” environment (for example, exposure to stressors or to different temperatures, nutrients availability, etc.), but also in the “internal” environment of cells and, in multicellular organisms, of tissues and organs. For example, energy requirements change according to the phase of cell cycle [1], vary between different tissues and organs [2], and are profoundly remodeled during physiological processes, like development and aging [3].

Aging is a multi-dimensional, multi-factorial process [4], in which the organism continuously adapts to the progressive accumulation of damage and the decay in maintenance systems that occur over time [5,6]. Age-related changes have been described at the epigenomic [7,8], transcriptomic [9,10], and proteomic level [11,12,13]. Regarding the proteome, the observed changes are, in part, related to the diminished efficiency of the proteostasis machinery, which leads to the accumulation of damaged, misfolded and/or aggregated proteins [14]. At the same time, a growing number of studies suggests that protein turnover decreases during aging [15,16] and that interventions that promote lifespan extension, like caloric restriction (CR) or the inhibition of target of rapamycin (TOR) pathway, imply a decrease in proteins synthesis [17,18]. As protein synthesis is the main ATP-consuming process in the cell [19], its decrease could have a beneficial effect during aging, as it would allow for optimizing energy homeostasis promoting its reallocation, for example, in favor of the repair of damaged molecules [16].

The key component in proteins synthesis is the ribosome, a complex ribonucleoprotein particle assembled in nucleoli. Ribosomal RNA genes (rRNA) are encoded by ribosomal DNA sequences (rDNA), which are highly evolutionary conserved and organized in higher eukaryotes as tandem head-to-tail repeats, distributed on the short arms of acrocentric chromosomes [20]. In mammals, each unit contains the 45S rRNA encoding sequence and a non-transcribed spacer [21,22]. Approximately 400 copies of rRNA genes are present in human cells, but only half of these copies are active in physiological conditions. The concerted activity of transcription factors and epigenetic marks, which are involved in the inactivation of the supplementary copies, regulate the transcription of rDNA units [23,24,25]. rRNA is the most abundant transcript of the cell and, as such, its transcription is highly energy consuming. In line with what is discussed above, rDNA transcription is modulated according to different environmental factors, like the regulators of cell cycle, growth factors, stressors, and nutrients, to guarantee energy homeostasis [26]. In particular, several studies reported that amino acid/glucose starvation and CR reduce rDNA transcription in yeast and mammal cells [26,27,28,29,30].

DNA methylation of rDNA locus is emerging as an important layer of regulation of rRNA synthesis, both in physiological and pathological conditions [31,32,33,34]. Data that were obtained in rodents suggest the occurrence of an increase in DNA methylation of the rDNA locus. Swisshelm et al. observed an age-related increase in the methylation of rDNA sequences (18S, 28S) in murine brain, liver, and spleen [35], and an age-dependent hypermethylation in rat tissues was also reported more recently [36,37]. In human whole blood, no reproducible associations of rDNA methylation levels and chronological age were observed, but the methylation status of a CpG site in the rDNA promoter correlated with cognitive performance and survival chance in the population investigated [37]. Recently, Wang and Lemos re-analyzed existing next-generations sequencing data and confirmed the age-associated hypermethylation of the rDNA locus in mice and developed a model of an epigenetic clock based on DNA methylation values at the ribosomal locus. This ‘ribosomal clock’ accurately estimates an individuals’ age, is responsive to interventions known to modulate life- and health-span, such as CR, and it can be applied to different species [38].

In this study, we aimed to examine the effects of CR on the age-associated epigenetic remodeling of rDNA locus. First of all, we quantitatively analyzed DNA methylation of three target regions in an rDNA unit in livers from rats at two, 10, and 18 months and evaluated the homologous regions in hepatocytes from human donors at different ages. Subsequently, we evaluated the effect of CR on rDNA methylation of rat livers at the three time points. Finally, we evaluated the observed profiles with respect to the expression of rRNA and of the *Rrn3* gene, which encodes for a master regulator of rDNA transcription.

## 2. Materials and Methods

### 2.1. Human Samples

Human hepatocytes were derived from a cohort of 32 subjects (age range: 0–82 years), which were composed of 22 patients undergoing partial or total hepatectomy and 10 organ donors. The cells were isolated, as reported previously [39], centrifuged at 80 g for 6 min., and then frozen as dry cell pellets at −80 °C until further use. Genomic DNA was extracted from the cell pellets using the AllPrep DNA/RNA/Protein Mini Kit (Qiagen, Hilden, Germany), following manufacturer’s instructions.

### 2.2. Rat Samples

A total of 24 Fisher 344 male rats, bred at the Department of Biomedical Sciences, University of Cagliari, were used. The Institutional Animal Care and Use Committee of the University of Cagliari, Italy, reviewed and approved all of the animal experiments. They were conducted in accordance with guidelines and regulations for good animal practice. Twelve rats were fed ad libidum (AL), while the remaining 12 were put under CR (corresponding to 70% of the AL ratio). The animals were killed at two months (4 AL and 4 CR rats), 10 months (4 AL and 4 CR rats), and 18 months (4 AL and 4 CR rats). After sacrifice, the liver tissues were isolated. Genomic DNA and total RNA were simultaneously purified from the same biopsy while using the AllPrep DNA/RNA/Protein Mini Kit (Qiagen, Hilden, Germany). The concentration of DNA and RNA samples was measured by Qubit dsDNA BR Assay and Qubit RNA BR Assay, respectively (Invitrogen, Carlsbad, CA, USA).

### 2.3. Quantitative DNA Methylation Analysis

The DNA methylation levels at rDNA genes were investigated with the Agena EpiTYPER system (Agena Biosciences, San Diego, CA, USA), which allows for quantifying DNA methylation of single CpG sites or of groups of adjacent CpG sites (CpG units). 500 ng of genomic DNA were bisulfite converted while using the EZ-96 DNA Methylation Kit (Zymo Research Corporation, Orange, CA, USA), according to the manufacturer’s instructions. 10 ng of bisulfite-treated DNA were amplified in PCR with specific primers reported in Appendix A. Amplicons were processed according to EpiTYPER protocol. CpG units with duplicated values in the EpiTYPER output were removed from further analysis.

In human samples, the methylation levels of three different regions within rDNA genes (promoter and 5′ regions of 18S and 28S sequences) were analyzed, as reported in a previous work [32]. The three amplicons included seven, 13, and nine CpG units, respectively.

In rat samples, DNA methylation levels were analyzed in the same three regions. We were able to measure methylation levels of five, 17, and three CpG units in the rat RiboProm, 18S and 28S regions, respectively. The Rat RiboPromoter amplicon overlaps with the region investigated in [37].

### 2.4. Gene Expression Analysis

The total RNA was treated with the Turbo DNA-free Kit (Thermo Fisher, Waltham, MA, USA) to remove contaminant genomic DNA. A total of 700 ng of RNA was then retrotranscribed to cDNA while using the High Capacity RNA to cDNA kit (Thermo Fisher, Waltham, MA, USA). TaqMan Gene Expression Assays (Thermo Fisher, Waltham, MA, USA) were used to evaluate the expression of Rn45s (Rn06667414) and of *Rrn3* (Rn01424527). Ppia (Rn00690933) was used as the internal control gene. cDNAs were denatured at 95 °C for 2′ before cyclic amplification (95 °C for 3′, 60 °C for 30′′ for 40 cycles). Quantitative real time PCR was carried out while using a Qiagen RotorGene instrument (Qiagen, Hilden, Germany). Each sample was run in duplicate. Gene expression fold change was calculated using the comparative DDCt method.

### 2.5. Statistical Analysis

All of the statistical analyses and graphics were produced while using R v3.5.1. Pearson’s correlation between DNA methylation levels at each CpG unit and age was calculated using the cor.test function; the correlation coefficient value and *p*-value were extracted from the results of this function. The analysis of the association between 5S and *Rrn3* expression with age was performed by robust linear regression using the *rlm* function implemented in the MASS R package, given the presence of biological outliers. Welch’s *t*-test was used to compare DNA methylation of each CpG site and expression levels between AL and CR rats. *p*-values < 0.05 were considered to be statistically significant.

## 3. Results

### 3.1. Age-Related Increase in DNA Methylation Levels of rDNA Genes in Rat and Human Liver

We measured DNA methylation of three target sequences (Materials and Methods) in liver tissues from young (two months), adult (10 months), and old (18 months) rats fed with AL diet, in order to investigate the age-related epigenetic remodeling at the rDNA locus [40]. All of the CpG sites in the three target regions showed a positive correlation between DNA methylation and age, which indicates a trend towards age-dependent hypermethylation for all the CpG units included in the target sequences. The correlation coefficient was statistically significant (*p*-value < 0.05) for three, seven, and two CpG units in the RiboProm, 18S and 28S amplicons, respectively (Figure 1).

Next, we investigated whether the age-associated hypermethylation of rDNA locus in liver is conserved in human hepatocytes. We observed a positive correlation between DNA methylation levels and age for all of the assessed CpG sites (Figure 2). This hypermethylation with advancing age was more marked in the RiboPromoter and in 28S amplicons, which included three and four CpG sites with statistically significant (*p*-value < 0.05) correlation between methylation and age. On the contrary, the CpG sites in the 18S amplicon only showed a trend to age-associated hypermethylation.

### 3.2. Impact of CR on the DNA Methylation Profiles of rDNA Locus during Aging

We evaluated the impact of CR on DNA methylation levels of rDNA locus in rat livers samples. At two and 10 months of CR, we observed a clear trend to an increase in methylation levels when compared to AL rats for several of the CpG units in the RiboPromoter, 18S and 28S amplicons (Figure 3). This trend reached the statistical significance threshold (*p*-value < 0.05) for the CpG 35 in the 18S amplicon. On the contrary, at 18 months, the methylation profiles of CR and AL rats were similar for all the three amplicons.

### 3.3. Impact of Aging and of CR on the Expression of the 45S Precursor and of the Rrn3 Gene.

Finally, we evaluated the expression of the 45S rRNA precursor in the same rat liver biopsies that were assessed for DNA methylation profiles. No evident changes in 45S expression were observed at the time points considered here, although the presence of an outlier at 18 months could potentially affect the analysis (Figure 4A). Therefore, we considered the expression of the ribosomal transcription factor *Rrn3*, which has been shown to mirror the rRNA expression pattern [41]. *Rrn3* expression was highly correlated with 45S expression (also in the outlier sample; Pearson’s correlation coefficient: 0.9, *p*-value < 0.01; Figure 4B), and robust linear regression showed a trend towards downregulation with aging, which did not reach statistical significance (Figure 4C).

We then compared the expression of the 45S and *Rrn3* transcripts in AL and CR rat livers at the three time points (Figure 5). At two months, we observed a trend towards downregulation of 45S and *Rrn3* expression in CR fed rats, which was marginally significant (*p*-value = 0.05) for *Rrn3*. At 10 and 18 months, on the contrary, we did not observe evident differences in 45S and *Rrn3* expression induced by CR.

## 4. Discussion

A large body of evidence indicates that CR promotes the extension of healthspan and lifespan in a wide range of organisms [42]. Likely, the beneficial effects of CR rely on multiple, highly connected intra- and inter-cellular pathways. These include the modulation of genomic stability [43], counteraction of age-related epigenetic alterations [44], promotion of autophagy and of the optimal balance between protein synthesis and degradation [45], maintenance of the stem cells niche [46], regulation of tissue microenvironment [47] and of gut microbiota [48], amongst others.

In this study, we evaluated the effects of CR on rDNA methylation in the context of the age-related epigenetic remodeling at this locus. Several evidence supports a central role of ribosome regulation during aging: ribosomal proteins are overrepresentated in studies investigating age-associated transcriptomic changes [49], and the association between small nucleoli and longevity has been reported [50]. We focused our attention on liver, a key metabolic organ that undergoes a relatively successful aging when compared to other body districts [51] and that is characterized by specific epigenomic and transcriptomic age-associated changes [52].

Previous studies reported an age-related increase in the methylation of rDNA promoter in mouse and rat liver [35,37]. We confirmed these results and showed that hypermethylation is not restricted to the rDNA promoter, but also occurs in the 5′ regions of 18S and 28S sequences, as previously shown [36]. Notably, all of the rodent tissues assessed in the different studies (liver, brain, spleen, heart, blood, kidney, testis, and sperm) display an age-related hypermethylation [35,36,37]. On the contrary, in humans, rDNA methylation changes with age have only been assessed so far in whole blood, where reproducible changes in methylation patterns have not been observed [37]. Here, we report for the first time that an increase in rDNA methylation also occurs in human hepatocytes. This result indicates that mammals differ for the tissue-specific regulation of rDNA methylation during aging, which suggests that a fine regulation of the epigenetic status of rDNA locus occurs in response to species- and tissue-specific requirements during aging. Unfortunately, the fact that rDNA locus is excluded from the design of the Illumina Infinium microarrays has prevented the analysis of rDNA methylation in tissues from large human cohorts. Future studies should specifically assess rDNA methylation remodeling during aging in different human tissues, by using pyrosequencing, EpiTYPER or targeted bisulfite sequencing, or by re-analysing reduced representative bisulfite sequencing (RRBS) or whole genome bisulfite sequencing data. Wang and Lemos recently used a similar approach, who re-analyzed a previously published dataset including whole blood DNA methylation of mice at different ages, as measured by RRBS [38]. They found several hundreds of CpG sites within the rDNA locus that underwent age-related hypermethylation, which is in line with our observations of a concordant trend towards an increase in methylation in three different regions of the rDNA unit.

We next considered the effects of CR on rDNA methylation. We found that this nutritional intervention caused hypermethylation of rDNA sequences, which was more pronounced after two and 10 months of dietary regiment, which indicates that, in our experimental settings, CR enhances the age-related rDNA methylation remodeling. This result seems to be specific for liver tissue and for the time points that we considered. Indeed, Wang and Lemos evaluated the changes of rDNA methylation in response to CR in mouse blood cells and found significant hypomethylation after the nutritional intervention. They analyzed in the same way a dataset of mouse liver upon CR, and found that rDNA hypomethylation occurred at 26 months, but not at five months of CR, when a trend towards rDNA hypermethylation was actually observed. Therefore, our experimental results are in line with the bioinformatic predictions that were achieved by Wang and Lemos.

At the same time, it should be noted that, in humans, rDNA hypermethylation has been associated to age-related conditions and pathologies, including Werner syndrome (WS) [53] and a decrease in cognitive performance and survival chance [37,54]. Tissue-specific hypermethylation was also described in brain samples from patients with mild cognitive impairment and Alzheimer’s disease [55]. Collectively, these data reinforce the idea that tissue-specific regulation of rDNA methylation occurs during aging, probably depending on the age-related microenvironment and the protein/energy homeostasis requirements, which differ across different body districts [16,56,57].

Finally, we evaluated whether the observed age- and CR-associated hypermethylation of rDNA locus is associated with changes in rRNA transcription. Differently from [37], we did not observe a significant decrease in the expression of the two transcripts. However, it should be observed that, in D’Aquila et al., the largest differences in liver pre-rRNA expression were observed between three and 28 weeks rats, while no changes were observed at later time points. We evaluated the expression of R*Rrn3* gene to get more insights on the transcriptional regulation of rDNA in our experimental settings. The Rrn3 protein has a well established function in regulating the RNA Polimerase I machinery in response to external stimuli, including stress and nutrients availability [58,59,60], and its expression has been shown to correlate with pre-rRNA expression in response to maternal diet [41]. We confirmed a good correlation between 45S and *Rrn3* expression and found a non-significant trend towards age-associated *Rrn3* downregulation. When considering the effects of CR on 45S and *Rrn3* expression, we found that, at two months of CR, the two transcripts tended to be downregulated when compared to age-matched controls, while no reproducible changes were observed at later time points. It should be noted that the effects of the epigenetic profile of rDNA locus could go beyond the transcriptional regulation of rRNA, as nucleolus function controls a wide range of additional processes, like genomic and nuclear stability, stress response, and regulation of cell cycle and growth [61].

Interestingly, our results are collectively in line with those that were reported in a recent study, which analyzed the effects of a low protein diet that was administered to female mice before implantation [41]. A decrease in total RNA content, an increase in rDNA methylation and a decrease in *Rrn3* expression levels were observed in liver fetal tissue of the offspring, and the overexpression of *Rrn3* in the human kidney cell line HEK293 induced an upregulation of 45S and a decrease in rDNA methylation. Additionally, in accordance with our data, smaller nucleoli have been found in the liver of mice undergoing CR for one month [50].

In this study, we focused on DNA methylation, but it is worth noting that other types of epigenetic modifications are likely to act in concert with DNA methylation to regulate rRNA expression and nucleolar functions during aging and CR. For example, DNA methylation profiles are strictly correlated with histone modifications at the rDNA locus [62,63], and active and inactive rRNA genes have different histone modification profiles [21,23]. A genome-wide remodeling of histone marks occurs during aging [64,65] and CR [66,67], thus future studies should specifically address the relationship between DNA methylation and histone modifications at the rDNA locus during aging and CR in liver tissues.

## 5. Conclusions

In conclusion, this study showed that age-related hypermethylation of rRNA occurs in rat and human livers, and that this process is enhanced by CR at two and 10 months in the liver of rats. Future studies should clarify whether the observed changes are beneficial during aging, according to the protein/energy requirements of the tissue context, and how the epigenetic regulation of rDNA locus is modulated in pathological conditions and upon pharmacological or nutritional interventions promoting healthy aging and longevity.

## Figures and Tables

**Figure 1 nutrients-12-00277-f001:**
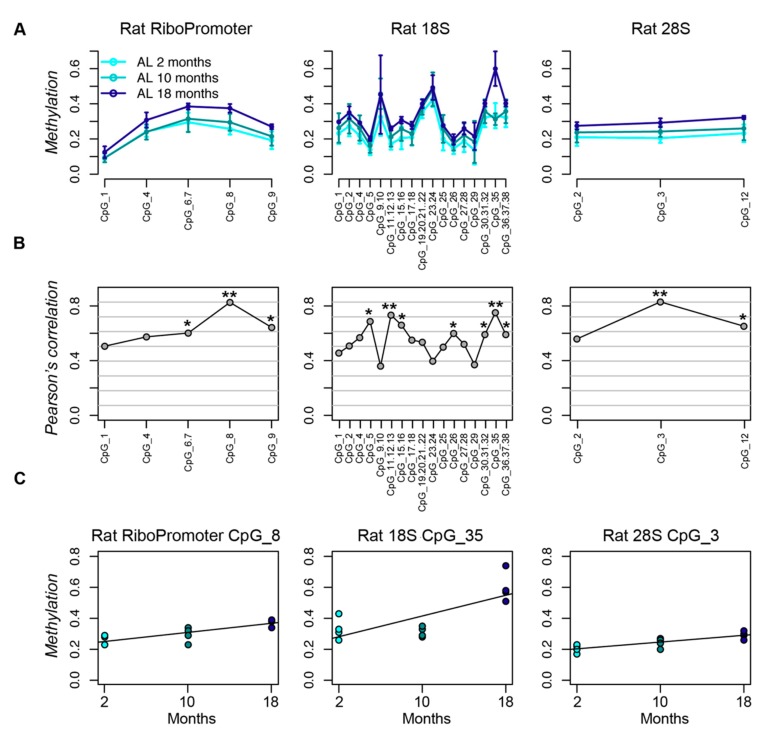
The rDNA locus undergoes hypermethylation during aging in rat liver. (**A**) Lineplots of DNA methylation profiles of the RiboPromoter, 18S and 28S amplicons in liver tissue from rats fed ad libitum for two, 10, and 18 months. For each CpG unit, mean methylation and standard deviation across four biological replicates for each time point are reported. (**B**) Coefficients of Pearson’s correlation between DNA methylation and age for each CpG site in the three target regions. (**C**) Scatterplots of DNA methylation values vs. age for the CpG unit most associated with age in each amplicon. *: *p*-value < 0.05; **: *p*-value < 0.01.

**Figure 2 nutrients-12-00277-f002:**
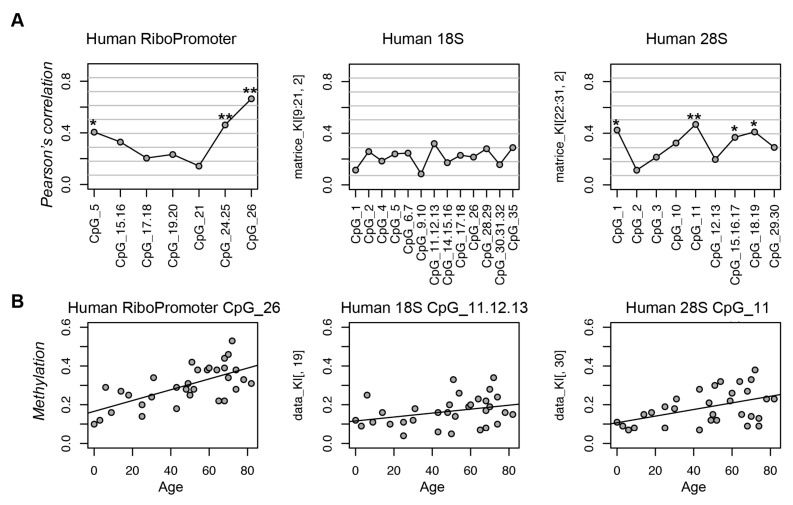
The rDNA locus undergoes hypermethylation during aging in human liver. (**A**) Coefficients of Pearson’s correlation between DNA methylation and age for each CpG site in the 3 target regions. (**B**) Scatterplots of DNA methylation values vs. age for the CpG unit most associated with age in each amplicon. *: *p*-value < 0.05; **: *p*-value < 0.01.

**Figure 3 nutrients-12-00277-f003:**
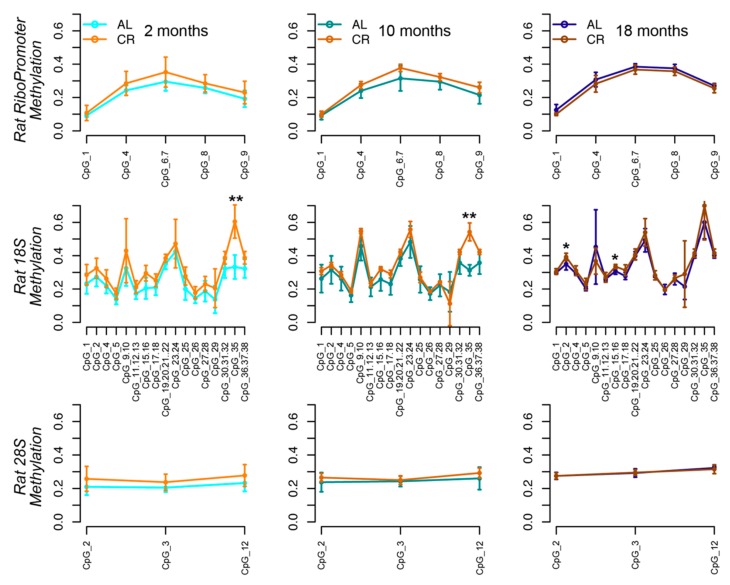
The rDNA locus undergoes hypermethylation in livers from caloric restriction (CR) fed rats. Lineplots of DNA methylation profiles of the RiboPromoter, 18S and 28S amplicons in liver tissue from rats fed ad libidum (AL) and under CR for two, 10, and 18 months. For each CpG unit, mean methylation and standard deviation across four biological replicates for each time point are reported.

**Figure 4 nutrients-12-00277-f004:**
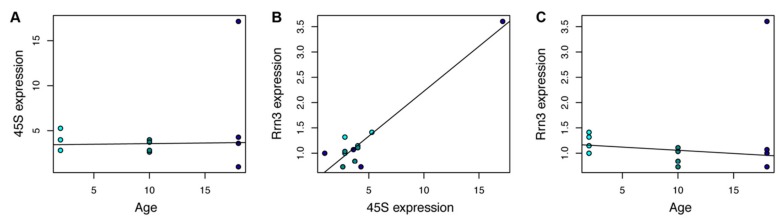
45S rRNA precursor and Rrn3 expression do not significantly change with aging in rat livers. (**A**) Scatterplot of 45S expression vs. age in rat livers. (**B**) Scatterplot of 45S expression vs. *Rrn3* expression in rat livers. Samples are colored according to age, using the colors used in panel (**A**). (**C**) Scatterplot of *Rrn3* expression vs. age in rat livers.

**Figure 5 nutrients-12-00277-f005:**
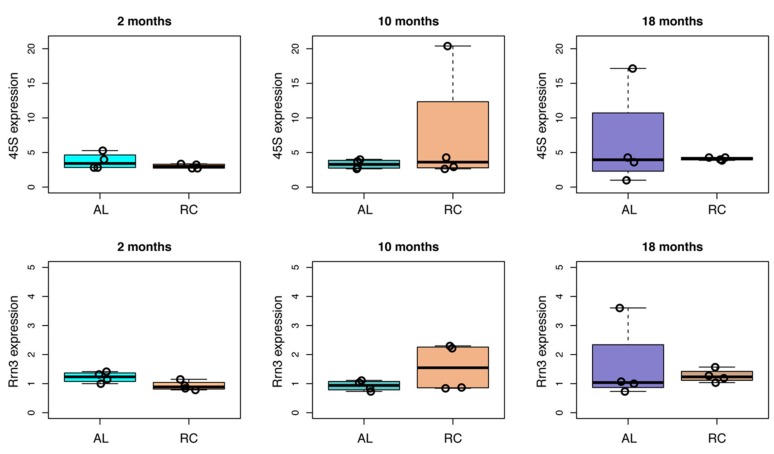
Effect of CR on the expression of 45S and of *Rrn3* in aging rat livers. For each time point (two, 10, and 18 months) the expression of 45S and *Rrn3* transcripts in AL and CR rat livers is reported as boxplots. Values from individual animals are reported as dots.

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
