# Peer review of "Aging and Caloric Restriction Modulate the DNA Methylation Profile of the Ribosomal RNA Locus in Human and Rat Liver"

_nutrients, 2020, doi:10.3390/nu12020277_

Round 1

Reviewer 1 Report

This is a novel study that now allows further exploration of caloric restriction in humans focused on aging and disease. My only criticism is how the Results section was written. A lot of the information in this section relates to the methodology and as such, should not be included in the Results section.

Please see my addition comments below:

Line 33: After “caloric restriction” abbreviate – “(CR)”

Line 68: “rRNA” 

Line 72: “rDNA” has not been defined.

Line 75: “In line with what is discussed above…”

Line 78: “amino acid”

Line 89: “…developed a model of an epigenetic clock based..”

Line 90-91: “..accurately estimates an individuals’ age and is responsive….”

Line 93: “…we aimed to examine the effects of caloric restriction….”

Line 95: “..in an rDNA unit..”

Line 98: “..observed profiles in respect to the expression..”

Line 98: “….and of the Rrn3 gene…”

Lines 149-154: This information should be in the Methods, but some of it has already been mentioned.

Line 168: “..isolated from human liver..”

Lines 167-171: This information should be in the Methods, but some of it has already been mentioned.

Lines 183-185: This information should be in the Methods, but some of it has already been mentioned.

Line 237: “heart”

Line 254: “..in line with our..”

Line 258: “canidis” – is this correct?

Line 273: Delete “a” before “CpG”

Line 298: “…effects of a low protein diet administered to female mice..”

Line 303: “…found in the liver of mice undergoing CR for 1 month.”

Line 305: “In conclusion, this study showed….”

Line 306: “…in the liver of rats.”

Author Response

Dear Reviewer,

Thank you for your recommendations; we have addressed them in the attached revised version of the manuscript, that you find in track changes form.

Please find below a point-by-point response to your comments:

“My only criticism is how the Results section was written. A lot of the information in this section relates to the methodology and as such, should not be included in the Results section”

The incipit of the 3 paragraphs has been modified and methodological information has been moved to the Materials and Methods section.

Please see my addition comments below”

All the corrections have been integrated in the text.

Reviewer 2 Report

The paper is actually written quite well. The authors are clearly versed in this topic and have thoroughly searched the literature. However, a few major comments to improve this:

The main concerns about the paper are:

150- At this point it should be mentioned the basis of selecting the 2-, 10- and 18-months old rats, the reader looses the rationale of selecting these aged rats.  155-Out of of 5, 17 and 3 CpG units in the rat RiboProm, 18S and 28S regions, there were only 3, 7 and 2 CpG units in the amplicons respectively which were statistically significant. Does it mean 7 out 17 CpG sites is a sign of positive correlation, and what happen to the other CpG sites? are they demethylated? 182-It would be very interesting to know the chromatin state at the different age points and during caloric restriction at the promoter region. Since the Histone modification and DNA methylation are often correlated, it would provide the much detailed picture at the locus, and would help us understand the mechanism of calorie restriction and aging. 209 & 290- What are the changes in the protein levels of Rrn3? do authors claim that there could be similar trend of protein levels Rrn3 as that of its transcripts. Authors need to show a western blot of the Rrn3 protein and correlate it with the transcription. The discussion needs to be written more concisely.

Minor points/spelling mistakes Please replace ''CpG units'' to ''CpG sites'' 59- write ''diminished'' instead of ''diminuished''. 120 & 122- Write 500ng instead of five hundred nanogram, and 10ng instead of ten nanogram. 254- ... ''in line'' instead of ''in linr'' Figures: The authors should provide better resolution figures, Figure 1 and Figure 3, it is hard to read through these figures.

Author Response

Dear Reviewer,

we have appreciated your comments and suggestions; we have addressed them in the attached revised version of the manuscript, that you find in track changes form.

Please find below a point-by-point response to your comments:

“150- At this point it should be mentioned the basis of selecting the 2-, 10- and 18-months old rats, the reader looses the rationale of selecting these aged rats.”

The incipit of section 3 has been modified as follows:

“In order to investigate the age-related epigenetic remodeling at the rDNA locus, we measured DNA methylation of 3 target sequences (Materials and Methods) in liver tissues from young (2 months), adult (10 months) and old (18 months) rats fed with AL diet [40].”

155-Out of of 5, 17 and 3 CpG units in the rat RiboProm, 18S and 28S regions, there were only 3, 7 and 2 CpG units in the amplicons respectively which were statistically significant. Does it mean 7 out 17 CpG sites is a sign of positive correlation, and what happen to the other CpG sites? are they demethylated? 1

All the CpG units in the 3 amplicons undergo hypermethylation with aging. We have clarified this in the text as follows:

“All the CpG sites in the 3 target regions showed a positive correlation between DNA methylation and age, which indicates a trend towards age-dependent hypermethylation for all the CpG units included in the target sequences. The correlation coefficient was statistically significant (p-value < 0.05) for 3, 7 and 2 CpG units in the RiboProm, 18S and 28S amplicons respectively (Figure 1).”

82-It would be very interesting to know the chromatin state at the different age points and during caloric restriction at the promoter region. Since the Histone modification and DNA methylation are often correlated, it would provide the much detailed picture at the locus, and would help us understand the mechanism of calorie restriction and aging.

The Reviewer’s suggestion is highly pertinent. Unfortunately, we do not have biospecimens and reagents to perform ChIP experiments within the revision deadline. However, we added in the Discussion a paragraph describing this point:

“In this study we focused on DNA methylation, but it is worth to note that other types of epigenetic modifications are likely to act in concert with DNA methylation to regulate rRNA expression and nucleolar functions during aging and CR. For example, DNA methylation profiles are strictly correlated with histone modifications at the rDNA locus [62,63] and active and inactive rRNA genes have different histone modification profiles [21,23]. A genome-wide remodeling of histone marks occurs during aging [64,65] and CR [66,67], thus future studies should specifically address the relationship between DNA methylation and histone modifications at the rDNA locus during aging and CR in liver tissues.”

209 & 290- What are the changes in the protein levels of Rrn3? do authors claim that there could be similar trend of protein levels Rrn3 as that of its transcripts. Authors need to show a western blot of the Rrn3 protein and correlate it with the transcription. The discussion needs to be written more concisely.

Unfortunately, again we do not have biospecimens and reagents to perform ChIP experiments within the revision deadline. Tissues were extracted by the AllPrep kit, but proteins are stored in Urea 8M and, on the basis of our experience, Western Blots not always perform well in these conditions. Discussion has been modified in order to make it more concise.

Minor points/spelling mistakes Please replace ''CpG units'' to ''CpG sites'' 59- write ''diminished'' instead of ''diminuished''. 120 & 122- Write 500ng instead of five hundred nanogram, and 10ng instead of ten nanogram. 254- ... ''in line'' instead of ''in linr'' Figures: The authors should provide better resolution figures, Figure 1 and Figure 3, it is hard to read through these figures.

We prefer to use the expression CpG units, as it is the nomenclature used in the EpiTYPER assay. For all the other corrections, we modified the text as suggested and improved the figures.

Round 2

Reviewer 1 Report

Congratulations on a very interesting study.

Reviewer 2 Report

The authors have revised the text of the manuscript, however it is still very confusing that they are unable to do ChIP and western blots in the respective . Surely the manuscript will be benefited by these experiments. The claim that authors do not have the samples, is not sufficient is not convincing.

I hope the authors will figure out to do these experiments to present a better picture of their data.